# Complete Chloroplast Genomes of 14 Subspecies of *D. glomerata*: Phylogenetic and Comparative Genomic Analyses

**DOI:** 10.3390/genes13091621

**Published:** 2022-09-09

**Authors:** Yongjuan Jiao, Guangyan Feng, Linkai Huang, Gang Nie, Zhou Li, Yan Peng, Dandan Li, Yanli Xiong, Zhangyi Hu, Xinquan Zhang

**Affiliations:** College of Grassland Science and Technology, Sichuan Agricultural University, Chengdu 611130, China

**Keywords:** chloroplast genome, comparative genomics, sequence divergence, RNA editing sites, phylogeny, divergence time, ploidy

## Abstract

Orchardgrass (*Dactylis glomerata* L.) is a species in the Gramineae family that is highly important economically and valued for its role in ecology. However, the phylogeny and taxonomy of *D. glomerata* are still controversial based on current morphological and molecular evidence. The study of chloroplast (cp) genomes has developed into a powerful tool to develop molecular markers for related species and reveal the relationships between plant evolution and phylogenetics. In this study, we conducted comparative genomic analyses and phylogenetic inferences on 14 cp genomes of *D. glomerata* originating from the Mediterranean and Eurasia. The genome size ranged from 134,375 bp to 134,993 bp and exhibited synteny of gene organization and order. A total of 129–131 genes were identified, including 85–87 protein coding genes, 38 tRNA genes and 8 rRNA genes. The cp sequences were highly conserved, and key sequence variations were detected at the junctions of inverted repeats (IRs)/small single–copy (SSC) regions. Moreover, nine highly variable regions were identified among the subspecies based on a sequence divergence analysis. A total of 285 RNA editing sites were detected that were relevant to 52 genes, where *rpoB* exhibited the most abundant RNA editing sites. The phylogenetic analysis revealed that all *Dactylis* subspecies clustered into a monophyletic group and most branches provided a high support bootstrap. The main divergence time of *D. glomerata* was dated to the Miocene era, and this could have been due to changes in the climate. These findings will provide useful insights for further studies on phylogeny, the identification of subspecies and the development of hypotheses for the evolutionary history of the genus *Dactylis* and of the Gramineae family.

## 1. Introduction

Orchardgrass (*Dactylis glomerata* L.) is a member of the Gramineous family [1]. The genus contains one species and numerous subspecies that have different ploidy levels [2]. Orchardgrass is native to Eurasia and northern Africa, although it has been introduced to nearly every continent and utilized as excellent cool–season forage for the livestock industry that requires forage [3]. In China, orchardgrass has already become an elite forage crop for mixed pasture construction and rocky desertification improvement in southwestern regions. It is highly valued economically and has a strong potential for utilization, in addition to its substantial ecological importance [4]. Based on ploidy, there are primarily three categories of the genus *Dactylis*: diploid (2n = 2x = 14), tetraploid (2n = 2x = 28) and hexaploid (2n = 2x = 42) [5]. Tetraploid populations are widely used in forage production, while hexaploid populations are rarely reported [6]. For a long time, taxonomists focused on the evolutionary relationship between orchardgrass diploids and tetraploids. Nevertheless, the absence of taxonomically diagnostic characters, and the high morphological similarity among the species, has made it difficult to classify the genus [7]. Although previous research has attempted to classify the *Dactylis* genus at the cytological and genetic levels, a unified standard for the taxonomy of the subspecies is still lacking. *Dactylis* classification has been repeatedly revised based on herbaria and field studies [8], cytology and genetics [9]. Molecular genetic analysis of partial DNA sequencing recently [2,5] changed clustering from three (a diploid and two tetraploid groups) to two (comprising, respectively, 17 diploids and six tetraploids). In one sense, although these studies have greatly improved the understanding of the taxonomy and phylogeny of the *Dactylis* genus, controversy about the phylogenetic status and interspecific phylogenetic relationship of the *Dactylis* genus remains.

The chloroplast (cp) is a key organelle with substantial functions in photosynthesis, carbon fixation, translation, and transcription [10,11]. Typically, a cp has a highly conserved genome sequence that contains 100–130 genes with a range of sizes from 120–170 kb in most land plants [12,13]. A cp genome usually has a covalently closed circular molecular structure that contains two inverted repeats (IRs) separated by a large single–copy (LSC) region and a small single–copy (SSC) region [14]. Compared with nuclear genomes, cp genomes are characterized by a highly conserved genome structure, a moderate substitution rate and uniparental inheritance, which have been widely used for plant phylogeny, and to identify species, estimate divergence and generate genetic markers [15,16]. In recent years, rapid advances in next–generation sequencing (NGS) technologies have enabled many studies with high–quality genomes with raw reads—such genome sequencing generates genome sequences much more rapidly and economically than using traditional Sanger sequencing [17,18]. Therefore, we have witnessed a greatly increased number of complete plant cp genomes in recent years. To date, more than 6500 complete cp genome sequences have become available in the National Center for Biotechnology Information (NCBI) since the first cp genome, that of tobacco (*Nicotiana tabacum*), was sequenced in 1986 [19].

However, there have been few studies on the comparative analysis of cp genomes in the *Dactylis* genus to date. In this study, we sequenced and assembled complete cp genomes of 14 subspecies of *D. glomerata* using next–generation sequencing. Subsequently, we analyzed the structural characteristics of the genome and identified the variant regions and RNA editing sites in the cp genomes. In addition, the divergence date was estimated, and phylogenetic relationships were reconstructed to assess the taxonomic positions of *Dactylis* species. This study aims to provide a reference for the taxonomy, phylogeny, and population genetics of *D. glomerata*. The research should facilitate the exploration and utilization of forage resources.

## 2. Materials and Methods

### 2.1. Plant Material and DNA Extraction

Fresh leaves of 14 subspecies of *D. glomerata* were collected from the Sichuan Agricultural University experimental greenhouse (30°42′ N, 103°51′ E; Chengdu, China). The lighting was set to 14 h/10 h (day/night) with a temperature of 22 °C/15 °C (day/night) (Appendix A). The total genomic DNA was extracted from 100 mg of fresh leaves using a Plant Genomic DNA Kit (Tiangen, Beijing, China).

### 2.2. Chloroplast Genome Sequencing, Assembly and Annotation

The fragmented DNA was prepared for libraries using a VAHTS Universal DNA Library Prep Kit for Illumina V3 (Jisi Huiyuan Biotechnologies Co., Ltd., Nanjing, China); the sequencing read length was PE150. The libraries that passed quality inspection were sequenced on an Illumina Nova Seq 6000 Platform (San Diego, CA, USA). The raw reads were filtered with a threshold of average quality <Q5 and a number of N > 5 using the fastp tool (version 0.20.0, https://github.com/OpenGene/fastp, accessed on 1 April 2022) [20]. The cp genomes of 14 *Dactylis* subspecies were assembled using the SPAdes pipeline (v3.10.1) [21]. To annotate the cp genome coding sequences (CDS), rRNA and tRNA were acquired using prodigal v2.6.3 (http://www.github.com/hyattpd/Prodigal, accessed on 1 April 2022) [22], hmmer v3.1b2 (http://www.hummer.org/, accessed on 1 April 2022) [23] and ARAGORN v1.2.38 (http://130.235.244.92/ARAGORN accessed on 1 April 2022) [24]. In addition, cp genomic data were extracted from NCBI using BLAST v2.6 (http://blast.ncbi.nlm.nih.gov/Blast.cqi, accessed on 1 April 2022) to align the assembled sequences [25]. The annotation results were then manually checked, and incorrect annotations were removed, along with redundant annotations. The boundaries of multiple exons were determined. Finally, circle maps of all the cp genomes were drawn using the program OGDRAW v1.1 [26].

### 2.3. Genome Comparison

To compare the structure of the cp genomes of all the *D. glomerata* subspecies, the borders between the IR and SC regions were analyzed. The program mVISTA was used to compare the cp genomes of 14 *Dactylis* subspecies, with the annotation of *D. aschersoniana* used as the reference [27]. Mauve (v2.3.1) software was used to analyze the homology and collinearity to align the genome [28].

### 2.4. Repeat Element Analysis and RNA Editing Identification

MISA (Microsatellite Identification Tool, v1.0) software was used to analyze the cpSSR using parameters 1–8 (single base repeat 8 times or more) [29]. The software Vmatch combined with a perl script was used to identify repeat sequences [30]. The parameters were as follows: minimum length = 30 bp, Hamming distance = 3, and the four identification forms were forward, palindromic, reverse, and complement. The editing sites of genes among the 14 subspecies of *D*. *glomerata* were predicted using PREP–CP with a cutoff value of 0.8 (http://prep.unl.edu/cgi-bin/cp-input.pl, accessed on 1 April 2022) [31].

### 2.5. Phylogenetic Analysis and the Estimation of Divergence Time

The cp genome sequences were used to construct a phylogenetic tree. The interspecies sequences were aligned by MAFFT software (v7.427–auto mode), and trimAl (v1.4. rev 15) was used to trim the well–aligned data [32]. Using RAxML software (v8.2.10) [33], a GTAGAMMA model with bootstrap analysis executed with 1000 replicates was used to construct the maximum likelihood evolutionary tree. Based on the phylogenetic tree obtained, mcmctree software in the paml (v4.9) package was used to construct the phylogenetic tree of divergence time, and the fossil time of species was searched from the timetree website (http://www.timetree.org/, accessed on 1 April 2022) [34]. Time calibration was based on fossil records of *Oryza sativa*, with a confidence range of 42–52 Mya. *Trapa* and *Corynocarpus* served as outgroups.

## 3. Results

### 3.1. Features of the Cp Genomes of the 14 Subspecies of D. glomerata

All the cp genomes of *D. glomerata* subspecies contained a typical quadripartite structure, comprising two IR (IRA and IRB) regions separated by a long single copy (LSC) and a small single copy (SSC) region. The length of the cp genomes of 14 subspecies of *D. glomerata* ranged from 134,375 bp (*D. glomerata* subsp. *lobata*) to 134,993 bp (*D. glomerata* subsp. *lusitanica*) (Figure 1). The lengths of LSC, SSC and IRs ranged from 79,753 to 79,773 bp, 12,246 to 12,276 bp and from 21,236 to 21,479 bp, respectively. The size of the LSC, SSC and IRs, as well as those of the whole cp genomes, were shorter than those of others in the two cp genomes of the subspecies *D. glomerata* subsp. *lobata* and *D. glomerata* subsp. *hispanica.1* (Appendix A). The total GC content was 38–38.44%, while the average *GC* contents of the LSC, SSC and IR regions were 36.32%, 32.86% and 43.94%, respectively (Appendix A). Moreover, a total of 129 genes were identified in the *D. glomerata* subspecies, excluding 131 genes identified in *D. glomerata* subsp. *hispanica.1.* The number of genes encoding the rRNA and tRNA was highly conserved, with eight genes encoding rRNA, and 38 genes encoding tRNA. The cp genomes of *D. glomerata* contained 83 protein–coding genes, while 85 protein–coding genes were identified in the cp genomes of *D. glomerata* subsp. *hispanica.1* (Appendix A). Eight tRNA genes (*trnH*–*GUG, trnI*–*CAU, trnL*–*CAA, trnV*–*GAC, trnI*–*GAU, trnA*–*UGC, trnR*–*ACG* and *trnN*–*GUU*), four rRNA (*rrn4.5, rrn5, rrn16* and *rrn23*), and seven protein–coding genes (*rps19, rpl2, rpl23, ndhB, rps7, rps12* and *rps15*) were duplicated in the IRs, excluding *D glomerata* subsp. *hispanica.1* (Figure 1). A putative gene of unknown function, *ycf15*, was present only in IRs in *D. glomerata* subsp. *hispanica.1* among 14 subspecies of *D. glomerata* (Appendix A). Interestingly, we found that a unique gene, *infA*, was distributed in all *Dactylis* subspecies. A total of 21 intron–containing genes were found in the cp genomes of all the subspecies, including 13 protein–coding genes and eight tRNA genes (Appendix A). Twelve protein–coding genes and eight tRNA genes contained one intron, and two genes (*ycf3* and *rps12*) contained two introns. Slight differences were found in the sizes of introns between different types of *D. glomerata* subspecies, as shown in Appendix A. In all the cp genomes of *D. glomerata* subspecies, the *trnK*–*UUU* intron, including *matK*, contained the longest introns (2497–2523 bp).

### 3.2. Comparative Genome Analysis

Fourteen cp genomes of the *D. glomerata* subspecies had highly similar sequences (Figure 2). The IR regions were more conserved than the SC regions. Among these cp genomes, the sequences in coding regions were almost identical or nearly identical. On the other hand, sequences in the noncoding regions were highly variable relative to the sequences in coding regions. The highly divergent regions were found in the intergenic spacers and introns, including regions of *rps19*–*psbA*, *psbM*–*petN*, *trnG*–*UCC*–*trnT*–*GGU*, *psaA*–*ycf3*, *rbcL*–*psal*, *psbE*–*petL*, *rps12*–*trnV*–*GAC*, *trnV*–*GAC*–*rps12* and *rpl2*–*rpl23*. These highly divergent regions have the potential to be used for discrimination or phylogeny investigations of the 14 subspecies. Furthermore, we checked the possible rearrangement events, which indicated that the genome structures and gene sequences were basically identical, and no gene rearrangement had occurred (Appendix A).

### 3.3. IR Contraction and Expansion

The contraction and expansion of the chloroplast IR regions appear to have a substantial role in the process of plant evolution, which is regarded as the main reason for the variation in genome size and gene quantity in different plants. Among the 14 subspecies of *D. glomerata*, *D. glomerata* subsp. *lusitanica* showed the longest cp genome (134,993 bp), with an IR of 21,472 bp, while the shortest cp genome sizes (134,375–134,376 bp) and IRs (21,236 bp) were detected in two subspecies, *D. glomerata* subsp. *lobata* and *D. glomerata* subsp. *hispanica.1* (Figure 3). We compared the IR/SC junctions of the 14 subspecies of *D. glomerata* and found that the IR/SSC junctions varied slightly. The location of *ndhF* was 109–110 downstream of the IRb–SSC junction.

The SSC–IRa junction was located within the coding region of *ndhH*, and the partial sequence of *ndhH* was 1012 bp within the SSC region, while only 987 bp of *ndhH* was found within the SSC region in the subspecies *D. glomerata* subsp. *lobata* and *D. glomerata* subsp. *hispanica.1*. The sequence length of *ndhH*, which was located in IRa, was 170 bp, except in *D. glomerata* subsp. *lobata* and *D. glomerata* subsp. *hispanica.1*. Correspondingly, *rps15* was located 332 bp upstream of the IRa/SSC junction in *D. glomerata* subsp. *lobata* and *D. glomerata* subsp. *hispanica.1*, which had a 25 bp difference from the other 12 subspecies.

### 3.4. Characterization of SSRs and Repeat Sequences

Six categories of SSRs were detected in the cp genomes of 14 subspecies of *D. glomerata* (Figure 4). The number of cpSSRs varied in all the subspecies and ranged from 181 (*D. glomerata* subsp. *ibizensis*) to 185 (*D. glomerata* subsp. *judaica*). Most of the SSRs were mononucleotide SSRs, which comprised 66.65% of the total SSRs, followed by trinucleotide (22.97%) and tetranucleotide repeats (6.60%) (Figure 4B). Moreover, the number of tetranucleotide repeats was higher than the number of pentanucleotide and dinucleotide repeats. Pentanucleotide repeats were found in the cp genomes of 12 subspecies of *D. glomerata* except for *D. glomerata* subsp. *hispanica.2* and *D. glomerata* subsp. *ibizensis*. Notably, the hexanucleotide repeats were exceedingly rare across the cp genomes of these subspecies, and only one was identified in the cp genome of *D. glomerata* subsp. *santai* (Figure 4A). Most of the SSRs were distributed in the LSC regions, followed by the IR and SSC regions (Figure 4C). In contrast to the exon and intron regions, most of the SSRs were detected in the intergenic regions (Figure 4D). Moreover, most of the SSRs were comprised of the repeat type A/T, rather than the repeat type G/C (Appendix A).

Interspersed repeated sequences are a type of repeat that differ from tandem repeats, which are distributed in a decentralized manner in the genome. Typically, interspersed repeats are classified into four types: forward, palindromic, reverse and complement repeats. In this study, we merely found palindromic repeats and forward repeats, which were similar to the number of interspersed repeats in all the subspecies (Figure 5A,B). The number of interspersed repeats varied from 38 to 44 in all the subspecies. *D. glomerata* subsp. *ibizensis* had the highest number of interspersed repeats, including 24 forward and 20 palindromic repeats. Although there were significant differences between the forward repeats (30–273 bp) and palindromic repeats (30–21,479 bp) in length, the size of most of the interspersed repeats was between 30–35 bp (Figure 5C,D).

### 3.5. RNA Editing

We identified 285 RNA editing sites associated with 52 genes in the 14 cp genomes of *D. glomerata* subspecies (Appendix A). Twelve types of RNA editing events were detected, including C to T, A to C, A to T, T to C, T to A, C to G, G to C, G to T, T to G, A to G, G to A, and C to A editing. In addition, all the editing events occurred in the region of proteincoding genes. In terms of the number of RNA editing sites of these genes, *rpoB* exhibited the most abundant RNA editing (15 sites), followed by *rpoC1* and *matk* (14 sites). *ndhF* had a value of 13. However, nine genes (*psbH, psbK, psbZ, rpl16, rpl22, rps8, ndhG, ndhI*, and *atpH*) had only one RNA editing site. The RNA editing of six genes (*atpA*–*1148, ndhF*–*62, ndhK*–*128, petB*–*611, rps8*–*3*, and *rpoA*–*527*) occurred in the cp genomes of all the subspecies. Although most of the editing events caused changes in the amino acids, there were 116 sites that did not alter any amino acids, including stop codons, among the subspecies. A total of 130 sites were edited in the third codon position, 79 sites were in the second codon position, and 74 sites were in the first codon position. Many of the editing events increased the hydrophobicity of amino acids, including conversions from hydrophilic amino acids to hydrophobic amino acids (57), from hydrophilic amino acids to hydrophilic amino acids (83), hydrophobic amino acids to hydrophilic amino acids (20), and hydrophobic amino acids to hydrophobic amino acids (107). Among the 169 edited amino acid sites, the largest proportion of changes was from Ser to Leu (13 sites), followed by Ser to Pro (8 sites), and Pro to Leu (5 sites). RNA editing in *D. glomerata* subsp. *glomerata* and *D. glomerata* subsp. *himalayensis* was completely consistent and contained 71 RNA editing sites that were involved in 27 genes. On the other hand, RNA editing of *D. glomerata* subsp. *ibizensis* and *D. glomerata* subsp. *hispanica.1* was different in many cases from those of the other subspecies, and 77 to 78 editing sites were identified in 34 genes of these two subspecies.

### 3.6. Phylogenetic Analyses

The clade composed of four *Triticum* species was the sister clade to the *D. glomerata* species and had BS of 100% (Figure 6). The *Dactylis* species was strongly supported as a monophyletic group (bootstrap support (BS) = 100%). These 14 subspecies of *D. glomerata* were divided into three groups. Group I contained three subspecies, group II contained 11 materials of 10 subspecies, and group III contained one subspecies. Although a few nodes still had lower BS values, most of the nodes of the ML tree had high BS values (> 80%). In Group II, it was clear that four *D. glomerata* subspecies—the diploid subspecies *aschersoniana*, the tetraploid subspecies *glomerata*, the tetraploid subspecies *woronowii*, and the diploid subspecies *judaica*–were the sister lineages of the diploid subspecies *himalayensis* and demonstrated higher BS (100%). The diploid subspecies *aschersoniana* showed the closest relationship with the tetraploid subspecies *glomerata* followed by the diploid subspecies *judaica* and the tetraploid subspecies *woronowii.* The diploid subspecies *judaica* was closely related to the tetraploid subspecies *woronowii*. The tetraploid subspecies *marina* showed a closer relationship with the evolutionary branches composed of the diploid subspecies *lusitanica* and *ibizensis. The* tetraploid subspecies *lobata* and *hispanica.1* are sister species, with BS of 100%.

### 3.7. Divergence Time

Serial divergence times from the crop plants to *D. glomerata* were estimated at 53.0985 to 29.5353 Mya (Figure 7). The divergence of the four *Triticum* species (*T. aestivum*, *T. macha*, *T. timopheevii*, and *T. monococcum* subsp. *monococcum*) lagged far behind that of *Dactylis glomerata*, which shared a common ancestor at 28.5353 Mya. Our study revealed climate change might have been a driver for the divergence of *D. glomerata*. Based on our results, *D. glomerata* originated in the early Oligocene era (28.5383 Mya) and gradually began to diversify after the mid–Miocene era (15.3985–7.4043 Mya). The divergence times of the tetraploid subspecies *marina* and the diploid subspecies *santi*, *lusitanica* and *ibizensis* (15.3985 to 13.2162 Mya), all of which originated from the west of the Mediterranean, were much earlier than those of the other remaining subspecies. Moreover, the divergence event of the diploid subspecies *himalayensis,* the diploid subspecies *aschersoniana*, the diploid subspecies *judaica*, and the tetraploid subspecies *glomerata* and *woronowii* occurred in 11.7017 Mya.

## 4. Discussion

### 4.1. The Features of Dactylis glomerata Subspecies Cp Genomes

In this study, we assembled and compared the characteristics of the cp genomes of 14 subspecies of *D. glomerata,* with sizes within the range of angiosperms [35]. In agreement with many higher plants, the cp genomes of all the *D. glomerata* subspecies were highly conserved and displayed a typical quadripartite molecular structure [36,37]. Upon comparing the cp genomes from 14 subspecies of *D. glomerata*, we found that the genome structures were relatively conserved, with no rearrangement occurring in gene organization, but there were some very significant differences in terms of size among the cp genomes of the analyzed *D. glomerata* subspecies. The overall genome, LSC, SSC, and IR regions of the *D. glomerata* subsp. *hispanica.1* and the *D. glomerata* subsp. *lobata* were shorter than those of the others, which might have been caused by the contraction and expansion of the IR and SSC boundary regions [37,38]. As such, the existing differences may provide insights into the unique differences defining Gramineae species and subspecies [39,40]. Though there were some differences in cp, the genome size among the 14 subspecies, the total GC content and the number of genes were similar, partially reflecting the conservation of the cp genome in angiosperms. In terms of the GC content of the cp genomes of the 14 subspecies of *D. glomerata*, there was higher GC content in the IR regions than in the LSC and SSC regions, and this could be the result of there being four rRNA genes in the IRs [41]. A high GC content has always been conducive to the stability of the genome structure, making mutation difficult [42]. Thus, the IR was the most conserved region in the cp genome of the 14 subspecies of *D. glomerata*.

The cp genomes of most angiosperms contain 74 protein–coding genes in their cp genomes; however, five other genes were found among certain species, with variation also being observed in the *D. glomerata* subspecies [43]. Interestingly, a translation initiation factor, *infA,* has been independently lost multiple times during the evolution of different land plants. However, it appears in all the subspecies in this study [43]. Thus, we deduced that the presence of *infA* is an ancestral condition in these subspecies. In addition, a gene with a high frequency of absence from the 14 *D. glomerata* subspecies was *ycf15*. The function of *ycf15* has been a concern in previous studies–its potential as a protein–coding gene in the cp genome of angiosperms has been questioned due to its high frequency as a pseudogene [44,45,46]. It is estimated that the *ycf15* gene has completely disappeared from about 29 terrestrial plant lineages [47,48,49,50,51,52,53,54]. However, comparative analysis revealed that this gene was found in the *D. glomerata* subsp. *hispanica.1* but that it was not present in other listed *Dactylis* subspecies. Based on these findings, parallel losses of particular genes have occurred over the course of *Dactylis* evolution. In this study, this finding increased our interest in studying the function and evolution of *ycf15* in angiosperms in more detail. Thus, the *ycf15* loss event may provide useful information for a more in–depth study of the evolutionary history of *Dactylis* or of Gramineae species.

### 4.2. Divergence of Active Regions

In order to explain the level of genome divergence, sequence identity among cp DNAs was detected. Though there was a high degree of similarity at the cp genomic scale, highly divergent regions were found, which included nine non–coding regions (*rps19*–*psbA*, *psbM*–*petN*, *trnG*–UCC–*trnT*–GGU, *psaA*–*ycf3*, *rbcL*–*psal*, *psbE*–*petL*, *rps12*–*trnV*–GAC, *trnV*–GAC–*rps12* and *rpl2*–*rpl23*), which may be highly useful for further studies of phylogenetic relationships, the identification of species, and population genetics. In general, the cp genomes of the 14 subspecies of *D. glomerata* showed lower divergence levels in their coding regions and in IRs than in their non–coding regions and SSC regions, and the IR regions were the most conserved. The higher GC content in IRs might partially explain the divergence of the conserved nature of the IR and SC regions [42].

IR is an important indicator for measuring the structural stability of cp genomes. Generally, a high number of rearrangements are detected in cp genomes lacking an IR, and these rearrangement events have also been reported in various terrestrial plant lineages [55]. However, the comparative analysis indicated that no rearrangement events took place in the cp genomes of the 14 subspecies of *D. glomerata* with IRs. The presence of an IR could prevent the occurrence of rearrangement events, resulting in a lower species rearrangement rate, which might account for the absence of rearrangement events detected in the 14 subspecies of *D. glomerata* [56]. In addition, large and complex repeat sequences in cp genomes are associated with genome rearrangement and stabilization; they provide important information for understanding the evolutionary history of plant species [57,58,59]. In this study, about 38 to 44 repeats in the cp genomes of the 14 subspecies of *D. glomerata* were detected, a relatively small number [60]. The size of the repeats was studied in the cp genomes that were sequenced and compared–we found that most of the repeats ranged in size from 30 to 35 bp. Although almost all the repeats were not large repeats (>100 bp), the repeats in *Dactylis* were more abundant than those in other angiosperm cp genomes, ranging between 7 and 13 [61,62,63]. Given the correlation between repeat sequences and rearrangements, high rearrangement rates are likely to be observed in cp genomes with a high frequency of large repeat sequences (>100 bp) [64]. However, no rearrangement events were found to have occurred in our 14 subspecies of *D. glomerata*, which might be due to a lack of large repeat sequences. It is, therefore, not surprising that rearrangement events were not observed in the 14 *D. glomerata* subspecies, which explains the relatively higher degree of stability and conservation among the different *Dactylis* subspecies.

### 4.3. IR Contraction and Expansion

Although the IR is the most conserved region in the cp genome, the border contraction and expansion of IRs are regarded as common evolutionary events in plants [65]. Overall, there was a close association between the IR length and genome size. For instance, the *D. glomerata* subsp. *lusitanica* possessed the largest cp genome (134,993 bp) and had a relatively long IR (21,472 bp), whereas the *D. glomerata* subsp. *lobata* and *D. glomerata* subsp. *hispanic**a.1* had the smallest cp genomes (134,375–134,376 bp) and the shortest IRs (21,236 bp). Analysis of the contraction and expansion of IRs from 14 subspecies of *D. glomerata* revealed that the gene distribution at the boundary of the four regions of the cp genome followed a similar rule. However, relatively independent characteristics were observed in the microstructure, such as the locations of *ndhH, ndhF* and *rps15.* In Gramineae, *ndhH* exists near both ends of the SSC region and can extend to the IRs, and it plays an important role in the stability of the IR/SSC junction [66]. As a typical cp genome structure, *ndhH* extends into the IRs in *Dactylis*, which may represent an ancestral symplesiomorphy in Gramineae [67]. Longer extension of *ndhH* into IRa unique to the monophyletic *D glomerata* subsp. *lobata* and subsp. *hispanica.1* cluster among other subspecies indicates that the genome size variation of the 14 subspecies of *D. glomerata* was mainly due to border shifts and variation in IRs [68]. Thus, we speculated that *ndhH* might play an important role in the cp evolution within *Dactylis*. These variations broaden our knowledge of the evolution and genomic structure of the cp genome of *Dactylis*.

### 4.4. Characterization of SSRs

SSRs are considered the results of slipped strand mispairing during DNA replication, which is usually observed in plant cp genomes [69]. The SSRs of plant cp genomes have been widely used for molecular markers for studying species genetic variations due to a high polymorphism rate at the species level [70]. In total, six types of SSRs were identified, and similar numbers of SSRs were detected in the 14 subspecies of *D. glomerata*. These cpSSRs appeared more frequently in the LSC region than in the SSC and IR regions, consistent with results of other Gramineae species [71]. Among the repeat units that were identified, the mononucleotide repeat (A/T) unit was the most abundant repeat unit, of which A or T repeats accounted for the majority. Our results support the concept that the SSRs in cp genomes are typically composed of short polyadenine (polyA) or polythymine (polyT) repeats and rarely contain tandem guanine (G) or cytosine (C) repeats [72]. Among these SSRs, hexanucleotide repeats showed different distribution patterns. Hexanucleotide repeats were the most common types in all the *Oryza* species and in certain *Phalaris arundinacea* [60,73], while hexanucleotide repeats were species–specific in the present study. An assessment of the SSR categories indicated that hexanucleotide repeats only existed in *D. glomerata* subsp. *santai.* Since the chloroplast is highly conserved in an–giosperms, cpSSRs can be transferred between species and genera [66]. So, the SSRs identified in the cp genome of *Dactylis* could be used as useful molecular markers to identify these *D. glomerata* subspecies and related species in future studies.

### 4.5. RNA Editing

As a crucial post–transcriptional RNA modification process, RNA editing appears in almost all land plants [74]. In the cp genomes of 14 subspecies of *D. glomerata*, most of the RNA editing events predicted were of C to T. In these 14 subspecies, the amino acid transformation from serine to leucine was the most common type. The dominance of hydrophilic serine to hydrophobic leucine substitution among non–synonymous RNA editing suggests the evolutionary conservation of RNA editing [75]. The RNA editing sites are often detected in the first or second base of codons, causing the conversion of hydrophilic amino acids to hydrophobic amino acids [76,77]. This transformation increases the hydrophobicity of proteins and enhances their stability [78]. In our present study, approximately 57 of the amino acid changes were from hydrophilicity to hydrophobicity, and the increase in amino acid hydrophobicity might facilitate the formation of core residues in proteins [79]. Thus, the structure formed by hydrophobic mutations in the protein kernel is more stable than that formed by hydrophilic mutations, which may ultimately affect the secondary structure and function of proteins and expand their genetic information [75,80].

However, abundant RNA editing sites were identified in the third codon among *D. glomerata* subspecies, which were consistent with some Gramineae plants [81,82]. The chloroplast RNA editing of *D. glomerata* subsp. *ibizensis* and *D. glomerata* subsp. *hispanica.1* varied more than those of the other *D. glomerata* subspecies, which may be due to changes in the environmental adaptability or species specificity during long–term evolution. Thus, some specific RNA editing sites were also identified in some of the subspecies, which provided useful information for the origin and evolution of the *Dactylis* genus based on in–sights from chloroplast RNA editing. To our knowledge, this is the first study to report RNA editing among *D. glomerata* subspecies, elaborating the evolution of 14 subspecies of *D. glomerata* with a focus on RNA editing, and laying the foundation for further studies of RNA editing mechanisms in *D. glomerata* and other types of plants.

### 4.6. Phylogeny Analysis and Divergence Time

Although the phylogenetic relationships among *Dactylis* taxa have already been extensively studied using morphological, cytological, isozyme, phenolic flavonoid, and molecular techniques, the classification of the *Dactylis genus* remains controversial [2,9,83,84,85]. In the present study, the phylogenetic tree, based on the cp genome, contains subspecies with different ploidy levels, and the high bootstrap support shows the capacity of complete cp genomes to enhance the phylogenetic resolutions during the evolution of *Dactylis*, which might represent useful information on the origin and evolution of *Dactylis* [86,87]. *Dactylis* is strongly supported as a monophyletic group and as the sister group of clades consisting of four species of *Triticeae*. We estimated that *Dactylis* and four *Triticeae* species diverged at 28.5353 Mya, which reinforces previous findings using genomics data (17.5–29.6 Mya) [82]. Diversification events in the lineages of *Dactylis* are primarily predicted to have gradually begun after the mid–Miocene era (15.3985–7.4043 Mya) [88]. It is suggested that there were huge climate changes in the climate after the mid–Miocene era, with the late–Miocene era being colder than the early Oligocene era [88]. This global cooling led to the contraction of tropical forests to lower latitudes, resulting in many open habitats, which presented opportunities for Gramineae plants [89]. As a type of cool grass, the adaptive radiation and diversity of *D. glomerata* subspecies were probably driven by climate change [82,90]. The tetraploid subspecies *marina* and the diploid subspecies *santai*, *lusitanica* and *ibizensis* from the western Mediterranean were the species observed during the early successive divergence of *Dactylis*, suggesting that the western Mediterranean might have been a diverse phylogenetic center in the past.

This study showed that most phylogenetic clades underwent adaptive radiation of the diploid subspecies followed by the tetraploid species. Obviously, most diploids have a much earlier origin than tetraploids, which is also supported by previous conclusions from studies on flavonoid variability [83,84,87]. Notably, the tetraploid subspecies *marina* clustered with the diploid subspecies *lusitanica* and *ibizensis*, indicating that one of these diploid subspecies, or their common ancestor, could be a parental species of tetraploid subspecies. The morphology of these subspecies also supports this hypothesis since the tetraploid subspecies *marina* possesses characteristic papillose epidermal cells that are very similar to those found in the diploid subspecies *ibizensis*, indicating that they might be derived from a common ancestor [8]. It has previously been suggested that the tetraploid subspecies *glomerata* may have evolved from the hybridization of *aschersoniana* (2n = 14) and *woronowii* (2n = 48) [83]. We found that it formed a clade with the diploid subspecies *aschersoniana* in the cp genome tree, supporting this diploid subspecies as one of its parents. Moreover, our results revealed that the tetraploid subspecies *woronowii* and the diploid subspecies *judaica* were clustered into a clade, indicating a close relationship between the two. Based on morphological studies, mainly of phenotypic characteristics, Stebbins and Zohary considered the tetraploid subspecies *woronowii* to be more similar to the xeromorphic diploid subspecies *judaica*, which has somewhat adapted to a dry summer climate and is considered to be of more recent origin [83]. Thus, these two subspecies may originate from a recent common diploid ancestor. In the present study, the diploid subspecies *himalayensis* forms a well–supported clade with the diploid subspecies *aschersoniana*, tetraploid subspecies *glomerata* and *woronowii*, and the diploid subspecies *judaica*, indicating that the five subspecies are closely related. In addition, the tetraploid subspecies *woronowii* is closely associated with the ancient Eurasian temperate forest flora (including the diploid subspecies *himalayensis* and *aschersoniana*) [83]. As Stebbins and Zohary note, the diploid subspecies *aschersoniana* and the tetraploid subspecies *woronowii* have similar flavonoid components, whereas the diploid subspecies *himalayensis* and *aschersoniana* have more primitive compounds than the tetraploid subspecies *woronowii* [84]. Obviously, the diploid subspecies *himalayensis* and *aschersoniana* originated earlier than the tetraploid subspecies *woronowii*. In this study, the diploid subspecies *himalayensis* was found to share a common ancestor with four other subspecies with different ploidy levels at 11.7017 Mya. The diploid subspecies *himalayensis* revealed similar phylogenetic positions when tetraploids were included, suggesting that the diploid subspecies *himalayensis*, or its ancestral lineage, may have been a parent of these subspecies. Phylogenomic analysis of the cp genomes will help to uncover the mysteries and controversies surrounding the phylogeny of *Dactylis* species. However, the sequencing data of the *Dactylis* subspecies used in this study were limited. Therefore, we recommend the inclusion of cp genome sequences of other subspecies of *D. glomerata* in future studies to help elucidate the evolution from *D**actylis*.

## 5. Conclusions

This is the first report of the whole cp genome of *D. glomerata* subspecies. The 14 subspecies of *D. glomerata* showed synteny of gene order and contained similar IR boundary regions in their cp genomes. Moreover, we also obtained important genetic information, including SSRs, repeat sequences, divergent hotspot regions, RNA editing sites, and divergence times associated with the relationships between the *D. glomerata* species and other Gramineae. Our results may supply insights to resolve taxonomic discrepancies and phylogenetic relationships within the *D. glomerata* species, accelerating the identification and utilization of forage resources.

## Figures and Tables

**Figure 1 genes-13-01621-f001:**
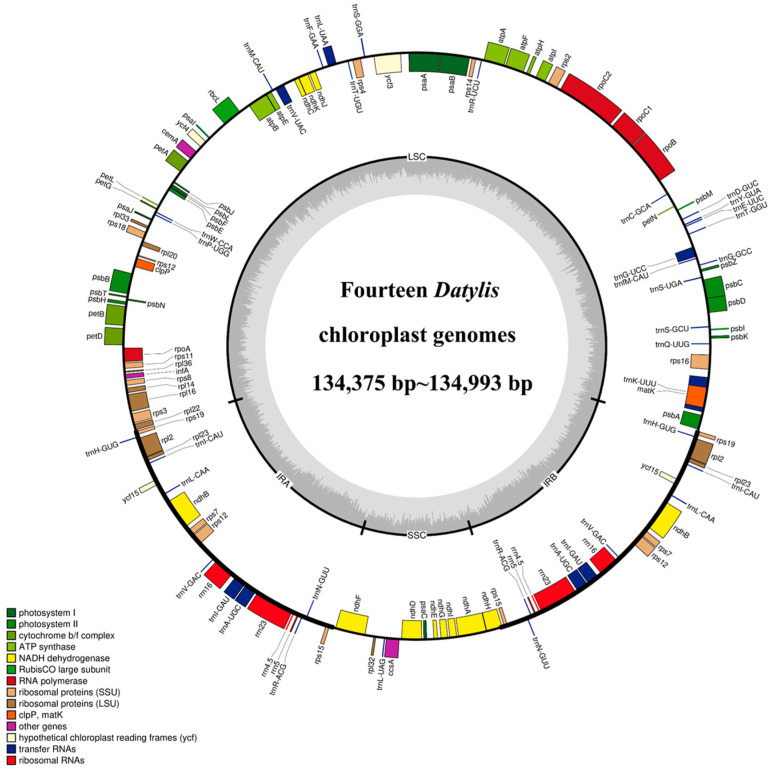
Representative cp genome of *Dactylis* subspecies. Genes drawn inside and outside of the circle are transcribed in clockwise and counter–clockwise directions, respectively. The colored bar indicates chloroplast gene groups. The dark gray bar graphs inner circle shows the GC content, and the light gray bar graphs show the AT content.

**Figure 2 genes-13-01621-f002:**
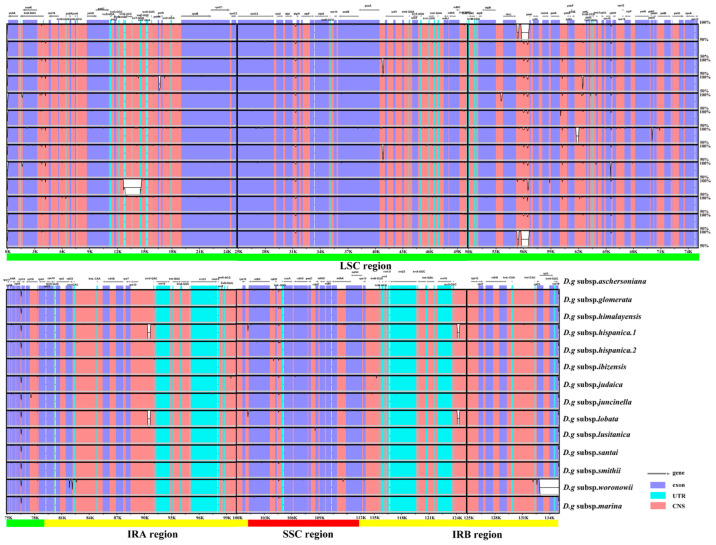
Sequence identity plot for cp genomes of *Dactylis* subspecies with *D.g* subsp. *aschersoniana*, as reference visualized by mVISTA. The gray arrows indicate the orientations of genes, the red blocks represent the intergenic region, the purple blocks represent exons, the blue blocks represent untranslated regions (UTRs), and the Y–axis represents the percent identity within 50–100%.

**Figure 3 genes-13-01621-f003:**
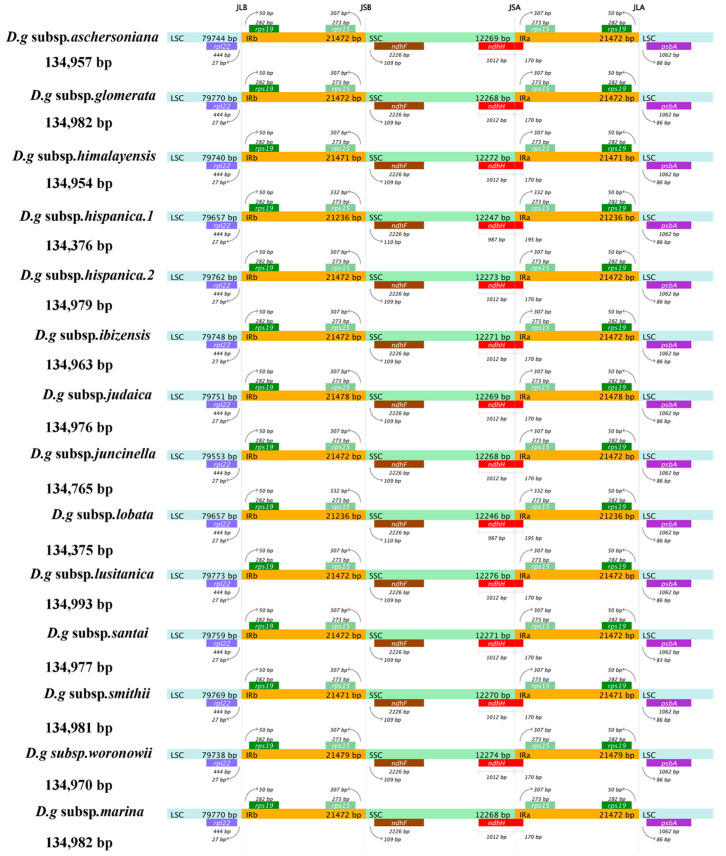
Comparison of the boundaries of LSC, SSC, and IR regions within cp genomes of 14 *D. glomerata* subspecies. Genes are denoted by bars, the gaps between the genes and the boundaries are indicated by the base lengths (bp). Extensions of the genes are indicated above the bars.

**Figure 4 genes-13-01621-f004:**
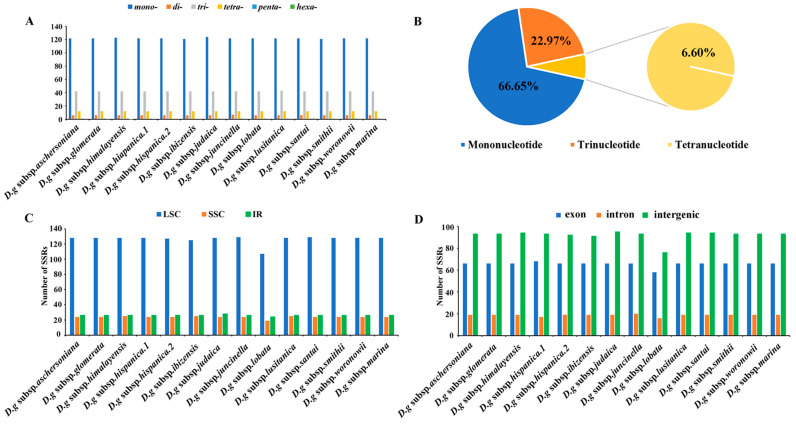
Analysis of simple sequence repeats (SSRs) in the cp genomes of 14 *D. glomerata* subspecies. (**A**) Number of different SSR types detected in cp genomes of *D. glomerata* subspecies; (**B**) The proportion of mononucleotide, trinucleotide and tetranucleotide SSRs; (**C**) Number of SSRs in LSC, SSC and IR regions; (**D**) Number of SSRs in exon, intron, and intergenic regions.

**Figure 5 genes-13-01621-f005:**
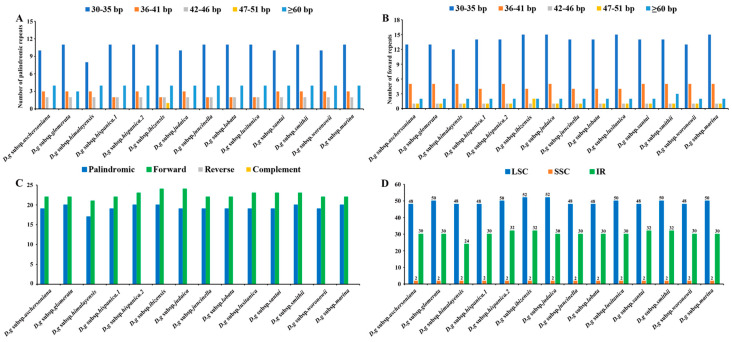
Analyses of repeated sequences in the cp genomes of 14 *D. glomerata* subspecies. (**A**) Number of palindromic repeats; (**B**) Number of forward repeats by length; (**C**) Number of different repeats; (**D**) Numbers of repeats in LSC, SSC and IR regions.

**Figure 6 genes-13-01621-f006:**
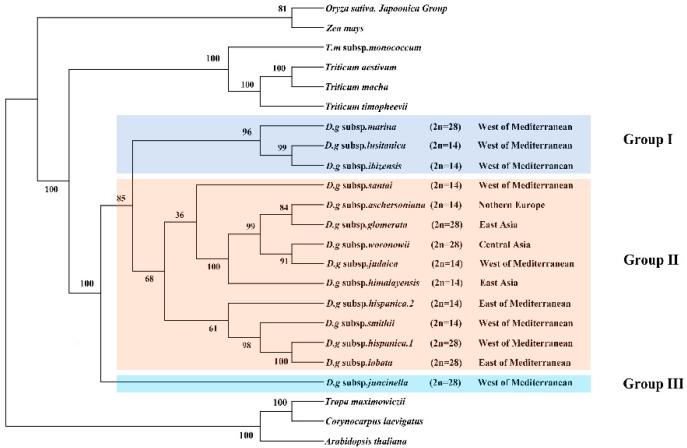
Phylogeny of *Dactylis* species inferred from complete chloroplast genome dataset. Phylogenetic tree constructed from *D. glomerata* subspecies and nine crop complete cp genome sequences using maximum likelihood (ML) method. Bootstrap values are marked above the branches. *Arabidopsis, Corynocarpus* and *Trapa* are used as the outgroups.

**Figure 7 genes-13-01621-f007:**
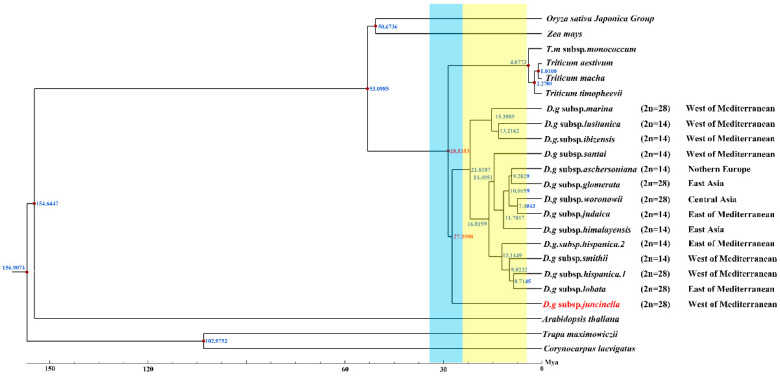
Divergence time of *D. glomerata* subspecies. Molecular dating of 14 *Dactylis* subspecies based on the cp genome sequences in cp genomes. Blue bar represents Oligocene period and yellow bar represents Miocene period.

## Data Availability

All the data pertaining to the present study has been included in the tables and/or figures in the manuscript. The authors are pleased to share analyzed/raw data and plant materials upon reasonable request. All the cp genome sequences of this study have been de–posited into the CNGB Sequence Archive (CNSA) of the China National Gene Bank Data Base (CNGBdb) with accession number CNP0002292, https://db.cngb.org/mycngbdb/submissions/pro–ject, accessed on 7 April 2022. The plant materials were provided by the Department of Forage Science, College of Grassland Science and Technology, Sichuan Agricultural University, Chengdu, China.

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
