# Peer review of "Complete Chloroplast Genomes of 14 Subspecies of D. glomerata: Phylogenetic and Comparative Genomic Analyses"

_genes, 2022, doi:10.3390/genes13091621_

Round 1

Reviewer 1 Report

Reports on Manuscript ID: genes-1895891

Type of manuscript: Article
Title: Insight into sequence variation and phylogenetic relationships of
Dactylis subspecies inferred from complete chloroplast genome
Authors: Yongjuan Jiao, Guangyan Feng, Linkai Huang, Gang Nie, Zhou Li, Yan
Peng, Dandan Li, Yanli Xiong, Zhangyi Hu, Xinquan Zhang *

Title: Authors write sequence variation – instead, they should write chloroplast genome comparison, and as all 14 subspecies belong to D. glomerata it should be clarified in the title.

I appreciate the annotation methods authors have used to nullify errors.

Authors have prepared many types of SSR marker charts but they should also focus on finding novel species-specific SSRs which really serve as a marker.

Here authors also state that Most D. glomerata subspecies diverged in the Mioceae (21.8187-7.4043 Mya) – which is also quite a long period to consider for subspecies level divergence. I suggest authors to reanalyze data for its validation.

In Ligands to the figures require English language corrections.

Other than this I also find minor grammar errors, please correct them.

Author Response

请参阅附件

Reviewer 2 Report

The manuscript entitled “Insight into sequence variation and phylogenetic relationships of Dactylis subspecies inferred from complete chloroplast genome” represents a major effort to evaluate the structural characteristics of genome and identification of the variant regions of cp genome of Dactylis glomerate to investigate the phylogenetic relationships and the taxonomic positions of Dactylis species. The major findings of this manuscript are useful for taxonomy, phylogeny, and population genetics of D. glomerata.

The introduction can be improved by adding some relevant literature. For example, it is recommended to address the following papers:

Arab MM, Brown PJ, Abdollahi-Arpanahi R, Sohrabi SS, Askari H, Aliniaeifard S, Mokhtassi-Bidgoli A, Mesgaran MB, Leslie CA, Marrano A, Neale DB, Vahdati K (2022) Genome-wide association analysis and pathway enrichment provide insights into the genetic basis of photosynthetic responses to drought stress in Persian walnut. Horticulture Research: 1-38.

Sadat-Hosseini M, Bakhtiarizadeh MR, Boroomand N, Tohidfar M, Vahdati, K. (2020). Combining independent de novo assemblies to optimize leaf transcriptome of Persian walnut. PloS One, 15(4), e0232005.

The methodology of this manuscript is clearly explained. Results are presented logically and coherently. Discussion of results is clear and adequate.

In conclusion, if my suggestions can be addressed successfully in a revision, then I believe the manuscript can be published in this journal.

In the abstract, it is not clear what kind of population did you use to sequence cp genome, and where they belong to? Where do they come from? However, this information can be found in the materials and methods.

In the Conclusion, the section includes a take-home message of your results about the phylogeny and taxonomy of subspecies of D. glomerate. What are the findings of your work for resolving controversies of the phylogeny of genus Dactylis?
